# Squeeze-MNet: Precise Skin Cancer Detection Model for Low Computing IoT Devices Using Transfer Learning

**DOI:** 10.3390/cancers15010012

**Published:** 2022-12-20

**Authors:** Rupali Kiran Shinde, Md. Shahinur Alam, Md. Biddut Hossain, Shariar Md Imtiaz, JoonHyun Kim, Anuja Anil Padwal, Nam Kim

**Affiliations:** 1Department of Information and Communication Engineering, Chungbuk National University, Cheongju 28644, Republic of Korea; 2VL2 Center, Gallaudet University, Washington, DC 20002, USA; 3Ashwini Rural Collage, Solapur 413006, India

**Keywords:** transfer learning, malignant, IoT, MobileNet, squeezed dataset, AUC-ROC, skin cancer detection, deep learning

## Abstract

**Simple Summary:**

Skin cancer is a life-threatening condition. It is difficult to diagnose in its early stages; therefore, we proposed an easy-to-use telemedicine device to tackle skin cancer without expert intervention. The deep learning model automatically detects skin cancer patches on lesions with a credit-card-sized device named Raspberry Pi and a small camera. This paper also presents a digital hair removal algorithm to enhance the quality of medical images for better analysis by medical experts and AI methods. Our method does not need an expert operator; even ordinary people can use it with the instruction manual. It will be useful for developing countries or remote places when there is a scarcity of oncologists.

**Abstract:**

Cancer remains a deadly disease. We developed a lightweight, accurate, general-purpose deep learning algorithm for skin cancer classification. Squeeze-MNet combines a Squeeze algorithm for digital hair removal during preprocessing and a MobileNet deep learning model with predefined weights. The Squeeze algorithm extracts important image features from the image, and the black-hat filter operation removes noise. The MobileNet model (with a dense neural network) was developed using the International Skin Imaging Collaboration (ISIC) dataset to fine-tune the model. The proposed model is lightweight; the prototype was tested on a Raspberry Pi 4 Internet of Things device with a Neo pixel 8-bit LED ring; a medical doctor validated the device. The average precision (AP) for benign and malignant diagnoses was 99.76% and 98.02%, respectively. Using our approach, the required dataset size decreased by 66%. The hair removal algorithm increased the accuracy of skin cancer detection to 99.36% with the ISIC dataset. The area under the receiver operating curve was 98.9%.

## 1. Introduction

In today’s world, cancer is a deadly disease. It is the 3rd most common cause of death among humans, with a 78% death rate at later stages. Skin cancer is an abnormal growth of skin cells that develops in the body due to sunlight and UV rays [1]. It quickly invades nearby tissues and spreads to other body parts if not seen at earlier stages. Early diagnosis of skin cancer is a foundation to improve the outcomes and is correlated with 99% overall survival (OS) [2,3]. This means there are higher chances of survival in the early stage. According to the Skin Cancer Foundation (SCF), there is an increase in skin cancer incidence globally [4]. More than 3 million cases will be detected worldwide in the year 2021.

The formal diagnosis method to detect cancer is visual inspection and biopsy. The primary visual examination includes the assistance of polarized light magnification via dermoscopy. A patient’s history, social habits, skin color, occupation, ethnicity, and exposure to the sun are the critical factors considered during examinations. The laboratory biopsies the suspected lesion of concern. This method is painful, times consuming, and expensive for doctors and patients. Without insurance, a skin biopsy costs $10 to $1000 [5]. There is an urgent need for skin cancer detection based on Artificial Intelligence (AI) to overcome the above problems.

Embedded devices have lower computing power, and there is a need for low memory-consuming AI models to work with them with better accuracy. Other researchers proposed great models, but detection time and memory requirements are higher. Therefore, we used the selected pretrained model to reduce training time. For better feature extraction, images have been preprocessed with a digital hair removal algorithm. This process reduced the computational cost of the model considerably.

Computer-aided diagnostic methods will transform healthcare and medicine. In terms of dermatology, various diagnostic models using medical images have been performed as well as clinicians [6]. Recently, deep learning has provided end-to-end solutions to detect COVID-19 infection, lung cancer, skin lesions, brain and breast tumors, stomach ulcers, and colon cancer; predict blood sugar levels and heart disease; and detect face masks [7,8,9,10,11,12]. Machine learning also contributes to enhancing the mathematical prediction of cancer cell spreading rate [13]. Ali proposes the novel use of sensory data to predict the patient’s length of stay in the hospital [14]. There are many deep learning models proposed by researchers, but very few are suitable for IoT devices. Most AI models require larger memory space and higher computational power for the best accuracy, but our model has optimal complexity and better accuracy. Imaging techniques have advanced rapidly; three-dimensional imaging systems, high-resolution digital cameras, and dermoscopes are used to obtain high-quality data from cancer patients worldwide. The International Skin Imaging Collaboration (ISIC) [15] created a digital dataset of skin lesions to facilitate the computer-aided design. The database includes images of melanomas and non-melanomas, as well as metadata. Some images are deliberately challenging for deep neural networks (DNNs) to interpret due to the presence of hair, ink marks, and rulers (Figure 1). Adequate data preprocessing is required.

Dr. Anuja Padwal (practicing in Solapur, India) has validated our device, stating that: “Patients have a high chance of survival in the early stages. Therefore, there is a need for affordable medical care. If we can commercialize this device, it will be used for primary analysis by a general physician or dermatologist”.

The key contributions of this study are as follows:We use a black-hat filter to efficiently clean the dataset and thus improve DNN accuracy efficiently. As our algorithm removes noise while reducing the size of the dataset, we call it the “Squeeze algorithm”;The architecture provides high accuracy (99.36%) and minimum loss of information with the transfer learning approach;The model was implemented and tested on an Internet of Things (IoT) device (Raspberry Pi 4) with a spy camera and NeoPixel 8-bit LED ring. The model is lightweight, precise, and optimized for IoT devices;The Squeeze-MNet outperforms the VGG16, MobileNetV2, and Inception V3 architectures.

The remainder of the work discussed in the paper is organized as follows: in Section 2, a review or some related work; Section 3 introduces some relevant theoretical methodology and DNN model architecture; Section 4 is an experimental setup with results and discussion; Section 5 is the conclusion of the proposed and future work.

## 2. Literature Review and Related Work

Most studies have used OpenCV and deep learning models for skin cancer classification. In early work, Friedman et al. proposed the ABCD (asymmetry, border irregularity, color, and diameter) abbreviation as a helpful mnemonic for nonprofessional and naive users aiming to identify common types of skin cancer melanomas at an early stage to allow early treatment [16]. ABCD helps us distinguish thin tumors and moles from benign pigmentation at an early stage. Later, ABCD was expanded to ABCDE [17], where E stands for “evolving”; the lesion is new or changing. Jensen et al. further expanded the ABCDE to ABCDEF [18], where F stands for a “funny-looking” mark. In [19], a genetic algorithm for extracting unique features from skin images was presented; the features were examined to determine if a disease was present [20].

Artificial neural networks (ANNs) are important classifiers [21,22]; their architectures have been modified to allow verification of nevi images and dataset classification. In [23], Josue presented an ANN with 99.23% accuracy when using Fourier spectral imaging. In another study, a convolutional neural network (CNN) detected skin cancer with high accuracy [24]. Meanwhile, Marwan combined a CNN with a novel regularizer to manage classification complexity; model accuracy exceeded 97%. Akhilesh et al. achieved a classification accuracy rate of 98% using a CNN and the color moments and textural features of a HAM10000 dataset with 7 different classes [25]. Image segmentation for feature extraction was combined with a generative adversarial network to improve classification [26]. An ANN classifier is used in this study. Lidia et al. [27] used a CNN with encoder and decoder architecture to remove hair via segmentation. Restored and original images, but not accuracy, were compared when evaluating performance.

Although deep learning is superior to hand-crafted feature representation, large annotated datasets are required, which professional oncologists lack the time to create. Thus, many studies [28,29,30,31,32,33] used transfer learning for skin disease classification to extract useful information from a previous dataset and apply it to a “raw target domain”; this obviates the need for expensive annotation of target data. Kessem et al. [30] used a pre-trained model and the GoogLeNet architecture to perform transfer learning using an ISIC 2019 dataset and successfully classified 8 classes of skin lesions using the Inception model; the accuracy was 94.2%. Hosny et al. used an AlexNet pre-trained model and the MED-NODE dataset for automated skin lesion classification. The model had a dropout layer and used the SoftMax activation function. An ImageNet dataset has been used to create pretrained, fine-tuned models, including MobileNet, InceptionV3, Resnet50, EfficientNet, and MobilenetV2. In [34], the accuracy of these models for skin cancer classification was 84%. Hari et al. [35] enhanced accuracy and precision to 90% and 89%, respectively, using the ResNet50 architecture. Even though deep learning models give the best results in cancer detection, less focus has been given to embedded-based systems. We train and test precise models on the Raspberry Pi platform with limited computation power.

In the 5G/6G internet era, the IoT is attracting considerable attention. During the COVID-19 pandemic, it became clear that healthcare IoT (HIoT) devices required artificial intelligence (AI). However, AI requires a large amount of memory and computing power, and IoT devices have memory constraints; thus, creating AI for HIoT devices is difficult. Therefore, we built a lightweight and efficient model trained on the ImageNet dataset to detect skin cancer.

## 3. Methodology

This section describes the Squeeze algorithm and model architecture (including a loss function). The input dataset (ISIC) is a publicly available data repository that undergoes regular preprocessing to enhance its quality [36]. Figure 2 depicts the data flow, including preprocessing, test and training datasets derivation, and model training and testing processes.

### 3.1. Duplicate Removal and Dataset Preprocessing

Duplicate and blurred images were manually removed, followed by dataset cleaning, application of a hair removal algorithm, and image augmentation; medical images are susceptible to noise. Then, we divided the dataset into training and test datasets (~80:20 ratio; Figure 3).

### 3.2. Hair Removal Using the Squeeze Algorithm

The removal of hair artifacts, which are common in dermatoscopic images, is essential and can be achieved using various complex segmentation techniques [27,37,38]. Our algorithm removes noise but not crucial information. The images are converted into grayscale and then subjected to thresholding using a black-hat filter. The hair mask contour yields a threshold image. OpenCV contains an “inpaint” function that restores a selected region in an image by reference to the neighboring area. We apply this function in the last stage to obtain uniform images with minimal information loss. Figure 4 shows the outputs of each process.

### 3.3. Augmentation

This augmentation process consists of image processing operations, such as rotation, flipping, shear, and scaling on the ISIC dataset. This process leads to better accuracy. It is useful when the dataset size is small; therefore, this process artificially increases the dataset size with the available images.

Zoom range = 0.15%,Shear range = 0.15°,Horizontal flip = True,Fill mode = nearest,Width shift range = 0.2°.

### 3.4. Model Architecture

We use a pretrained MobileNet model with predefined weights. Transfer learning saves time; existing (biased) weights are applied without sacrificing previously learned features. The head model accepts the outputs of the base model, i.e., the flattened and dense layers, and employs the Leaky Rectifier Linear Unit (ReLU) activation function to reduce the risk of overfitting. The “flatten” layer converts all two-dimensional arrays into a single long vector. The neural network uses the sigmoid activation function. The detailed architecture is shown in Figure 5. The sizes of the dense layers are 64, 32, and 2. There are two classification classes, such that either the binary cross-entropy or log loss function is optimal. The model compares the predicted probabilities to the actual class outputs (0 or 1) and applies penalties according to the difference between the actual and expected values [39]. Equation (1) is the loss function:(1)$$\mathrm{Log}\;\mathrm{loss}=\frac{1}{N}\sum_{1}^{N}-\left(yi*\log\left(pi\right)+\left(1-yi\right)*\log\left(1-pi\right)\right)$$

Here, pi is the probability of the “Malignant” class, and (1 − pi) is the probability of the “Benign” class.

## 4. Result and Discussion

### 4.1. Experimental Setup

The experiment was performed using an Intel Core i5-7500 3.40GHz processor running Windows 10 (32 GB of RAM, NVIDIA GeForce GTX 10050Ti graphical processor). The IoT device was a Raspberry Pi 4 microprocessor with a 64-Gb SD card. A spy camera and NeoPixel ring were attached to the camera port and main board. The ring ensures that photographs/videos taken at any time (day or night) are clear. The ring GND, 5V, and D1 pins were connected to pins 1, 6, and 12 of the Raspberry Pi, respectively. The device dimensions are 9 × 6.3 × 3.5 cm^3^. Figure 6 provides images of the device.

### 4.2. Hair Removal Algorithm

The algorithm significantly improved model accuracy and precision. Learning curves were plotted with and without hair removal. In the absence of preprocessing, the ISIC dataset was 162 MB, but this was dramatically reduced to 36 MB after processing. The peak signal-to-noise ratio (PSNR) and mean square root error (MSE) were 38.95 and 8.26, respectively; the reconstructed images were of high quality. The algorithm increased accuracy and reduced the training time by filtering irrelevant features; training focused only on important image regions. Figure 7 shows the random noise before and after processing; noise, but not important information, was removed.

Figure 7 shows that, after hair removal, the image is uniformly dense, but of lower intensity; black (score of 0) hairs were removed without losing critical information.

### 4.3. Squeeze-MNet Model Analysis

Standard metrics were calculated in this study, i.e., accuracy (Equation (2)), specificity (Equation (3)), sensitivity (Equation (4)), precision (Equation (5)), the false alarm rate (Equation (6)), and the area under the receiver operating characteristic curve (AUC-ROC) (Equation (7)):(2)$$Accuracy=\frac{TP+TN}{TP+TN+FN+FP}$$
(3)$$Specificity=\frac{TN}{TN+FP}$$
(4)$$Sensitivity=\frac{TP}{TP+FN}$$
(5)$$Precision=\frac{TP}{TP+FP}$$
(6)$$False\;Alarm=\frac{FN}{TP+FN}$$
(7)$$\mathrm{AUC}=\int_{0}^{1}f\left(x\right)dx$$

Table 1 compares our model to other pretrained models used for transfer learning. For training, there were 50 epochs and a batch size of 16 in all cases. An IoT model must be very accurate, but small in size; Table 1 shows Squeeze-MNet was optimal in terms of accuracy, training time, and the AUROC. The training time factor affects the speed of the detection; when training time and total extracted features are higher, the model is heavy (that is, needs huge memory to run the model), e.g., VGG-16 and Xception. When extracted features or training time is lower, the model gives less accuracy, such as with MobilenetV3Small mode. The proposed model gives the best tradeoff of size and accuracy. Figure 8 shows that the true-positive rate increased over time. ROC curve (Appendix A, Point 1) construction is necessary to evaluate an unbalanced dataset. The red line in Figure 8a shows the behavior of the untrained model, and the blue line denotes the accuracy after learning. The AUC-ROC measures model performance and ranges from 0 to 1. An ideal model has an AURC-ROC of 1; that of our model was 0.989. In Figure 8a, the red line is the performance of the model without knowledge, and the blue line is the intelligence gained with each epoch. This is about gaining logic; hence, we named it a logistic skill. AUC is equivalent to the probability that a randomly chosen positive instance is ranked higher than a randomly chosen negative instance. The overall error rate (1-accuracy) is 0.64, which is a combination of false positive and false negative. Both values are mentioned in the confusion matrix in Figure 8b. Our system does not perform well with false positive values. In the medical field, false positive is stressful, and false negative is fatal to patients. AI experts and researchers need to fix this issue.

The confusion matrix in Figure 8b shows the actual and predicted labels; the false-positive rate was lower when our model was tested using the validation dataset. Our model showed high accuracy and precision.

The learning curve (Appendix A, Point 2) shows that accuracy increased after each epoch because the model was fine-tuned (Appendix A, Point 3), and Leaky ReLU (Appendix A, Point 4) activation after each dense layer prevented overfitting (Appendix A, Point 5) and underfitting (Appendix A, Point 6). The Leaky version of ReLU allows only a small gradient to pass. Figure 9 shows the accuracy and loss by epoch. All the optimization work has been done by the head network mounted on the MobileNet deep learning model, as shown in Figure 5. Hyperparameter tuning is also responsible for the robust model. It is explained in Section 4.4.

### 4.4. Hyper-Parameters Tuning

We performed hyperparameter tuning; Table 2 provides details of the best-performing optimizer, decay constant, number of dense layers, and learning rate. We avoided overfitting and underfitting and focused on predictive accuracy. Hyperparameter optimization significantly improved model accuracy. The optimal hyperparameters were as follows:Optimizers: AdamLearning Rate: 0.001Weight Decay Values: 0.001Dense Layers Level: 3

When the model is evaluated to check false alarms from confusion matrix values, it gives 0.08%, which is great for medical applications. This means the model predicts the lowest number of false positive cases. The sensitivity and specificity from the confusion matrix are 95.2% and 96%, respectively. Our model is outperforming in accuracy, lightweight in terms of memory, and fast due to lower computations. Therefore, it is the most suitable model for skin cancer detection on IoT devices.

The deep learning model gives consistent performance when trained and tested on a good dataset. If the input image is not high quality, predictions can be wrong. The current error rate is lower, and it does not depend on the testing dataset size. In real life, the prevalence of positives is much lower, but the system does not have a memory to store previous results and gain knowledge from tested images. Therefore, it will not affect the efficiency of the system.

### 4.5. Comparative Study

We compared our model with traditional methods and deep AI-based models in terms of accuracy, precision, IoT compatibility, AUROCs, and methodologies (Table 3).

## 5. Conclusions

This study presented the novel Squeeze-MNet model for removing hair from dermoscopic images and classifying skin cancers using embedded systems, such as Raspberry Pi 4. Black-hat filtering during masking removes noise. A MobileNet model with a dense network architecture was extremely accurate when combined with the Squeeze algorithm (99.36%). A lightweight model is required for low-power devices; accuracy must be excellent, and the model size and training time should be small/low. Our model is precise and accurate. We optimized the hyperparameters and varied the optimizers, learning rates, and weighted decay values. We objectively compared the performance of our model with other models; Squeeze-MNet outperformed all other models. A digital hair removal algorithm has to be used before object detection, which can give a microsecond delay in detection. The major limitation of the system is that specificity and sensitivity are still lower than accuracy. For practical application, they have to be higher than current values. If the model is trained with a high-definition image dataset, we will overcome this issue. Another limitation is that the model only detects skin cancer, but cannot detect the type of cancer for further treatment. Currently, Squeeze-MNet is compatible with Raspberry Pi-based devices; in the future, we plan to make it compatible with Jetson Nano and the Google Coral Board. Fog and cloud computing may allow for AI implementation. We also plan to produce a deep learning-based cloud computing platform that minimizes computational costs at the edge. Although our system detects skin cancer, warts, acne, pimples, and eczema are also of concern. Ultimately, we aim to identify 35 skin diseases with our system, which will use AI to help dermatologists.

## Figures and Tables

**Figure 1 cancers-15-00012-f001:**
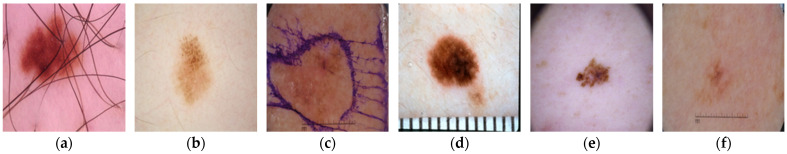
Challenging skin lesions within the dataset: (**a**) hair artifact, (**b**) low contrast, (**c**) ink marker, (**d**) ruler marker, (**e**) dark corner, (**f**) low illumination.

**Figure 2 cancers-15-00012-f002:**
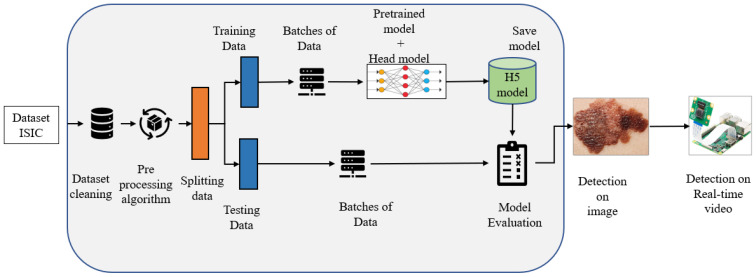
Flow diagram of the Squeeze-MNet model.

**Figure 3 cancers-15-00012-f003:**
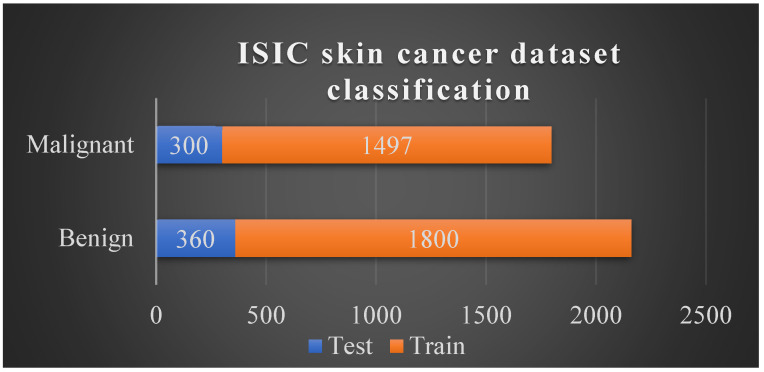
Dataset Distribution of data across the training and test datasets.

**Figure 4 cancers-15-00012-f004:**
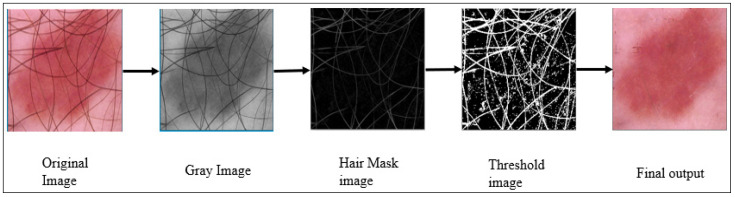
Images produced by the Squeeze algorithm during hair removal.

**Figure 5 cancers-15-00012-f005:**
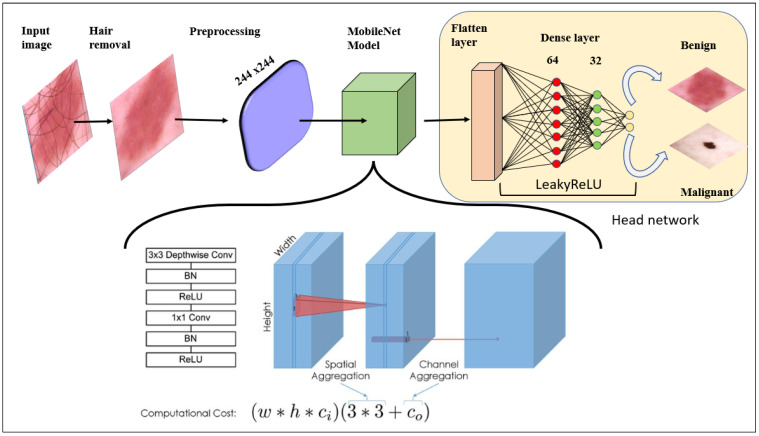
Model architecture.

**Figure 6 cancers-15-00012-f006:**
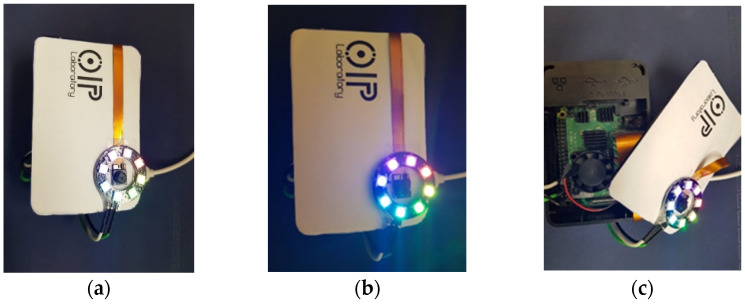
Images of the Device (**a**) front view, (**b**) device under low light, (**c**) internal view.

**Figure 7 cancers-15-00012-f007:**
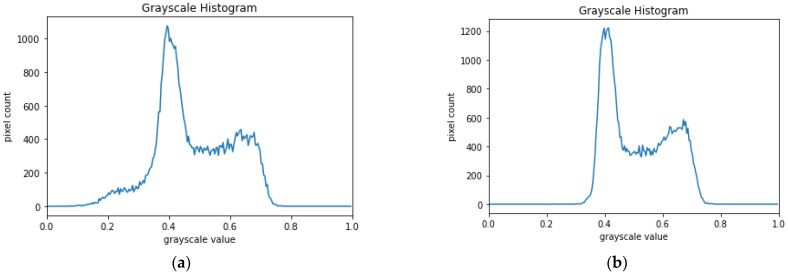
Histograms (pixel intensity vs. frequency) obtained from the image (**a**) before and (**b**) after digital hair removal.

**Figure 8 cancers-15-00012-f008:**
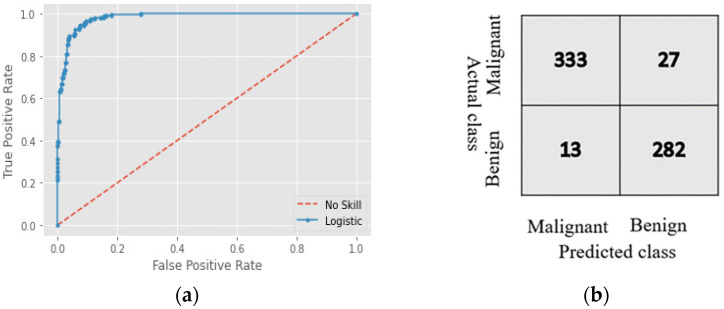
(**a**) ROC curve and (**b**) confusion matrix of the proposed model.

**Figure 9 cancers-15-00012-f009:**
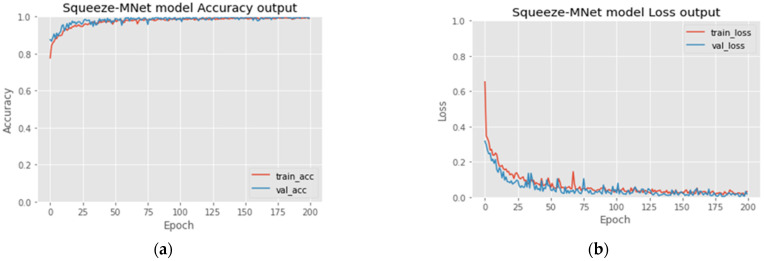
(**a**) Accuracy and (**b**) Loss per epoch.

**Table 1 cancers-15-00012-t001:** Comparative study with other pretrained models.

Model	Accuracy (%)	Average Precision (AP)	Recall	Training Time (s)	Model Size (KB)	Total Parameter	ROC-AUC (%)
MobileNetV2	85	89	87	1637.3	56,139	6,273,202	0.937
VGG-16	85	88	86	17,048.34	76,385	16,321,458	0.94
InceptionV3	80	83	88	5272.85	1,24,095	25,080,722	0.908
InceptionResNetV2	84	84	87	8945.21	2,42,301	56,795,474	0.929
Xception	82.2	84	84	6391.65	157,050	27,285,146	0.51
MobileNet	88	90	88	1885.62	50,439	6,441,266	0.949
MobileNetV3Small	74	82	90	1521.64	7074	1,596,642	0.829
MobilenetV3Large	76	79	86	1854.94	17,859	4,309,490	0.844
Squeeze-MNet(Proposed)	99.36	98	99	2271.60	50,439	6,441,266	0.989

**Table 2 cancers-15-00012-t002:** Effects of the changing hyperparameters on the Squeeze-MNet performance.

Optimizer	Learning Rate	Weight Decay Value	Dense Layer Level	Time/ Epoch (s)	Accuracy	F1-Score	Recall
Adam	0.001	0.001	3	804	99.56	98	99
Adam	0.001	0.01	3	809	98.01	98.05	97.00
Adam	0.01	0.01	3	801	65.42	95.45	96.02
SGD	0.001	0.001	3	802	97.07	92.30	99.01
SGD	0.001	0.01	3	806	95.23	89.11	97.08
SGD	0.01	0.01	3	800	92.48	88.70	96.12
RMSprop	0.001	0.001	3	815	98.96	97.08	96.99
RMSprop	0.001	0.01	3	802	95.40	92.49	93.00
RMSprop	0.01	0.01	3	806	93.23	91.09	91.89

**Table 3 cancers-15-00012-t003:** Comparison of our model with traditional and AI methods.

Author	Method	Accuracy (%)	ROC-AUC	Dataset	IoT Compatible
Nilkamal et al. [40]	ABCD	90	-	-	yes
Dang et al. [41]	ABCD	96.6	-	ISIC	Yes
Bandic et al. [42]	ABCDE	81.82	-	121 skin lesions	Yes
Kumar et al. [43]	ANN	97.4	-	HAM10000 & PH2	Yes
López-Leyva et al. [24]	ANN	99.23	97	Edinburgh Dermofit Library	Yes
Adekanmi et al. [44]	FCN-Densenet	98	99	HAM10000	No
Pham et al. [45]	EfficientNetB4-CLF	89.97	-	CIFAR-10	No
Silvana et al. [46]	IR camera & segmentation	91.5	-	400 images	Yes
Andre et al. [6]	CNN-PA	72.1	91	Edinburgh Dermofit Library	Yes
Uzama et al. [1]	GLCM-SVM	95	-	20 images	Yes
Bogdan et al. [47]	Deep uncertainty Estimation for skin cancer	98	-	ISIC	
Rehan et al. [48]	K-mean-CNN	97.9 & 97.4	-	DermIS & DermQuest	Yes
Saleh et al. [49]	YOLOv4	98.9	-	ISIC 2018&2016	No
Parvathaneri et al. [50]	MobilenetV2-LSTM	90.72	-	HAM10000	Yes
Marwan [24]	CNN	97.5	93	ISIC	Yes
Proposed	Squeeze-MNet	99.56	98.4	ISIC	Yes

## Data Availability

The data presented in this study are available on request from the corresponding author.

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
