# Peer review of "Squeeze-MNet: Precise Skin Cancer Detection Model for Low Computing IoT Devices Using Transfer Learning"

_cancers, 2022, doi:10.3390/cancers15010012_

Round 1

Reviewer 1 Report

This is a well-written paper describing a technological advance in an important area of public health and preventative medicine. The authors are commended for developing their ideas within a framework that will make the diagnostic system available in most of the world.  I particularly liked the fact that they demonstrated the low hardware requirements by generating data using their algorithm/classifier running on a very simple, low-cost device.

Specific, minor errors
Line 91: "then hand-crafted" should be "than hand-crafted"
Figure 2: In the flow diagram, the words below the orange operator should be "splitting data" not "spitting data".

In Figure 8 panel (a) the legend is not useful.  The concept of skill is not explained anywhere in the paper and in the context of medical diagnoses, the TPP=FPP diagonal is not usually referred to as the No Skill line (of course, technically, the definition is accurate, but will likely not be familiar to readers). What does "logistic" mean in this context?  As far as I can see the only place the word appears in the paper is in reference to the activation function in the ANN; why is it used to label the ROC curve?  In panel (b) the numbers in the 4 quadrants of the confusion matrix are unreadable.  Make the font larger and apply bold format to make the numbers stand out.  With the numbers readable, the color scale background isn't really needed, especially since on the scale used, only 3 values are needed. A simpler gray-scale background with fewer categories would be simpler and more relevant to the context.

Lines 238-239. This sentence needs to be rewritten. It is not currently in grammatical English.

General comments

Overall, the paper gives a clear narrative about the development of diagnostic system and the level of performance it achieves.  It is difficult in a paper of this type to find the balance between including enough technical detail to allow readers to understand the methods used, and over-filling the paper with technicalities that obscure the main message.  In this case I think the authors have done a reasonable job with this, but there are a few places where I think more detail is needed, and one or two methodological points that need to be addressed.

Throughout the paper there is quite a bit of technical vocabulary about machine learning, classifiers, and computational aspects of the development of the model.  Mostly these concepts are not described (for example, the phrase "learning curves" is used without any explanation about what a learning curve is, but this is just one example of several).  I was content to read the paper without a full understanding of what all these things are, but some readers will find this frustrating.  I think the authors should think about who their intended audience is.  If it is computer scientists and modelers, then the technical jargon does not need to be explained.  If the authors want their paper to be read and appreciated by epidemiologists and quantitative scientists working in public health, a glossary of terms in supplemental material would be a real benefit.

I found the section on model performance to be weak.  AUROC is not a very useful summary of performance.  Of course, if it's close to 1 then overall the model must be making few errors, but with overall accuracy levels in excess of 0.95 already available in other diagnostic systems, I would expect a more detailed examination of the comparative error rates for false positive and false negative diagnoses and a fuller discussion of the consequences of each of these types of error.

The precision (PPV) and false alarm values are dependent on the prevalence of true positives in the dataset (unlike specificity and sensitivity) but there is no acknowledgement of this in the paper, or any discussion of how these values would be expected to change when the device is used in practice, where the prevalence of positives may be much lower than in the database.

The model comparison in Table 1 is really interesting but also frustrating.  Recall is (again) a useful concept in machine learning but not widely used in the diagnostics literature; the same goes for F-1 score. Either include definitions for these terms in the paper, or add a glossary in supplemental material.

The text on lines 216-218 needs to be unpacked/expanded a little bit to help readers understand how what they are looking at supports the claims that are made.  I was frustrated throughout by the claims that the model tuning approaches avoided over-fitting.  This is an important issue for out-of-sample accuracy but it's glossed over in the paper. We have to rely on the assumption that the approaches the authors mention are robust in avoiding over-fitting, but if we are not familiar with them then it's not easy to be sure.  A brief explanation of how optimization is achieved without over-fitting would be useful. It's a key point and worth explaining in some detail.

Author Response

Reviewer 1

We would like to thank you reviewer for your valuable comments and suggestions. The answers to each comment are as given below.

---------------------------------------------------------------------------------------------------------------------

This is a well-written paper describing a technological advance in an important area of public health and preventative medicine. The authors are commended for developing their ideas within a framework that will make the diagnostic system available in most of the world.  I particularly liked the fact that they demonstrated the low hardware requirements by generating data using their algorithm/classifier running on a very simple, low-cost device.

Specific, minor errors
Line 91: "then hand-crafted" should be "than hand-crafted"
Figure 2: In the flow diagram, the words below the orange operator should be "splitting data" not "spitting data".

Response: As the reviewer mentioned line 91 and figure 2 was typing mistake. It has been corrected in the manuscript.

Action taken: (Corrected text)

Although deep learning is superior than hand-crafted feature representation, large annotated datasets are required, which professional oncologists lack the time to create.

In Figure 8 panel (a) the legend is not useful.  The concept of skill is not explained anywhere in the paper and in the context of medical diagnoses, the TPP=FPP diagonal is not usually referred to as the No Skill line (of course, technically, the definition is accurate, but will likely not be familiar to readers). What does "logistic" mean in this context?  As far as I can see the only place the word appears in the paper is in reference to the activation function in the ANN; why is it used to label the ROC curve?  In panel (b) the numbers in the 4 quadrants of the confusion matrix are unreadable.  Make the font larger and apply bold format to make the numbers stand out.  With the numbers readable, the color scale background isn't needed, especially since on the scale used, only 3 values are needed. A simpler gray-scale background with fewer categories would be simpler and more relevant to the context.

Lines 238-239. This sentence needs to be rewritten. It is not currently in grammatical English.

Response: In figure 8(a) logistics is used as logical gain by a model with each epoch. No skill refers to the model with zero skills. Details of the learning curve are added in the manuscript.

Action taken:(Added text:))

 In figure 8(a) red line is the performance of the model without knowledge and the blue line is the intelligence gained with each epoch. It is about gaining logic hence we name it a logistic skill.

Figure 8(b) has been changed as per suggestions and the updated figure has been shown below.

Line 238-239 has been rewritten as shown below.

When the model is evaluated to check false alarms from confusion matrix values it gives 0.08 % which is great for medical applications. The sensitivity and specificity from the confusion matrix are 95.2% and 96% respectively. Our model is outperforming in accuracy, lightweight in terms of memory, and fast due to lower computations. Therefore, it is the best suitable model for skin cancer detection on IoT devices.

General comments

Comment #1) Overall, the paper gives a clear narrative about the development of diagnostic system and the level of performance it achieves.  It is difficult in a paper of this type to find the balance between including enough technical detail to allow readers to understand the methods used, and over-filling the paper with technicalities that obscure the main message.  In this case I think the authors have done a reasonable job with this, but there are a few places where I think more detail is needed, and one or two methodological points that need to be addressed.

  • Throughout the paper there is quite a bit of technical vocabulary about machine learning, classifiers, and computational aspects of the development of the model.  Mostly these concepts are not described (for example, the phrase "learning curves" is used without any explanation about what a learning curve is, but this is just one example of several).  I was content to read the paper without a full understanding of what all these things are, but some readers will find this frustrating.  I think the authors should think about who their intended audience is.  If it is computer scientists and modelers, then the technical jargon does not need to be explained.  If the authors want their paper to be read and appreciated by epidemiologists and quantitative scientists working in public health, a glossary of terms in supplemental material would be a real benefit.

Response: It is a good suggestion therefore to understand the technical terms we have added appendix A to the manuscript. It is showing all the details of technical terminology for non-technical readers.

This paper is written by a Deep learning expert. Here on our team, we have a few medical experts. Therefore to make understanding easy we have explained technical terms in simplified words. Along with this simple summary and introduction sections have been expanded to understand the research objective and need of the study.

Action taken :  (text added)

Appendix A:

  1. A receiver operating characteristic curve, or ROC curve, is a graphical plot that illustrates the diagnostic ability of a binary classifier system as its discrimination threshold is varied. The area under the curve is called AUC-ROC.
  2. Learning curves are plots that show changes in learning performance over time in terms of experience. Learning curves of model performance on the train and validation datasets can be used to diagnose an underfit, overfit, or well-fit model. Learning curves of model performance can be used to diagnose whether the train or validation datasets are not relatively representative of the problem domain.
  3. The rectified linear activation functionor ReLU for short is a piecewise linear function that will output the input directly if it is positive, otherwise, it will output zero. It has become the default activation function for many types of neural networks because a model that uses it is easier to train and often achieves better performance.
  4. One way to increase performance even further is to train (or "fine-tune") the weights of the top layers of the pre-trained model alongside the training of the classifier you added. The training process will force the weights to be tuned from generic feature maps to features associated specifically with the dataset.
  5. Underfittingmeans that your model makes accurate, but initially incorrect predictions. In this case, train error is large and val/test error is large
  6. Overfittingmeans that your model makes not accurate predictions. In this case, train error is very small and val/test error is large.

Comment #2)I found the section on model performance to be weak.  AUROC is not a very useful summary of performance.  Of course, if it's close to 1 then overall the model must be making few errors, but with overall accuracy levels in excess of 0.95 already available in other diagnostic systems, I would expect a more detailed examination of the comparative error rates for false positive and false negative diagnoses and a fuller discussion of the consequences of each of these types of error.

Response: We agree with the reviewer that for medical applications, the AUROC curve is not considered an important parameter in many medical applications but it is a must in some cases where a false positive rate is important. The ROC curve is useful when the dataset is not balanced and confusion matric does not show an actual false positive rate. The benign and malignant classes do not have the same number of images. Therefore we checked performance with the ROC curve.

The ROC curve shows the trade-off between sensitivity (or TPR) and specificity (1 – FPR). Classifiers that give curves closer to the top-left corner indicate better performance. As a baseline, a random classifier is expected to give points lying along the diagonal (FPR = TPR).

The ROC does not depend on the class distribution. This makes it useful for evaluating classifiers and predicting rare events such as diseases or disasters. In contrast, evaluating performance using accuracy (TP +TN)/(TP + TN + FN + FP) would favor classifiers that always predict a negative outcome for rare events.

AUC is equivalent to the probability that a randomly chosen positive instance is ranked higher than a randomly chosen negative instance, i.e. it is equivalent to the two-sample Wilcoxon rank-sum statistic. A classifier with a high AUC can occasionally score worse in a specific region than another classifier with a lower AUC. But in practice, the AUC performs well as a general measure of predictive accuracy.

The available accuracy in medical diagnosis is 0.95 but it needs expert human resources and a huge cost for screening.  

The proposed system is accurate and economical to develop commercially. Along with that remote places and poor people do not have access to the best medical facilities. Our device can be used for them for early detection. If the result show some positive result respective person can connect to an oncologist for further treatment.

Action taken : (Text added in the manuscript)

In figure 8(a) red line is the performance of the model without knowledge and the blue line is the intelligence gained with each epoch. It is about gaining logic hence we name as a logistic skill. AUC is equivalent to the probability that a randomly chosen positive instance is ranked higher than a randomly chosen negative instance. The overall error rate (1- accuracy) is 0.64; It is a combination of false positive and false negative. Both values are mentioned in the confusion matrix in figure 8(b). Our system does not perform well with false positive values. In the medical field, false positive is stressful and false negative is fatal to patients. AI experts and researchers need to fix this issue.

Added text between 276-281 lines as below:

When the model is evaluated to check false alarms with confusion matrix values it gives 0.08% which is great for medical applications. The sensitivity and specificity from the confusion matrix are 95.2% and 96% respectively.

Comment #3) The precision (PPV) and false alarm values are dependent on the prevalence of true positives in the dataset (unlike specificity and sensitivity) but there is no acknowledgement of this in the paper, or any discussion of how these values would be expected to change when the device is used in practice, where the prevalence of positives may be much lower than in the database.

Response: The reviewer highlights the performance of the device when used practically. The deep learning model gives constant performance when trained and tested on a good dataset. If the input image is not high-quality prediction can be wrong. The current error rate is lower and it does not depend on the testing dataset size. In real life prevalence of positives is much lower but the system doesn’t have a memory to store previous results and gain knowledge from tested images, therefore, it will not affect the efficiency of the system. The author presented the evaluation factor in table 1 like accuracy and precision. The ROC curve is showing a minimum false positive rate after model training. The ideal AUC is 1 and our model is close to 1; hence minimum false positive rate.

False alarms include false positives and false negatives. Each application needs a good tradeoff between both.  Scientists have to be careful when they make decisions. They try to minimize errors and collect additional information or perform a test multiple times. This is difficult because reducing one type of error often increases the other type of error. Based on the consequences of their decision, one type of error may be preferable to the other.

Further sensitivity, specificity, and a false alarm are calculated with a confusion matrix.

Sensitivity = TP/(TP+FN) = 95.2

Specifity = TN / (TN+FP) = 96.00

False alarm = FN /(TP+FN) = 0.08

All of the above values are showing the lowest false alarm rate. It means it can be good for medical applications.

Action taken : (Text has been added on page 8)

The overall error rate (1- accuracy) is 0.64; It is a combination of false positive and false negative. Both values are mentioned in the confusion matrix in figure 8(b). Our system does not perform well with false positive values. In the medical field, false positive is stressful and false negative is fatal to patients. AI experts and researchers need to fix this issue.

(Text has been added on page 9)

When the model is evaluated to check false alarms it is 0.0.8% when calculated with present confusion matrix values which is great for medical applications. The sensitivity and specificity from the confusion matrix are 95.2% and 96% respectively.

The deep learning model gives constant performance when trained and tested on a good dataset. If the input image is not high-quality prediction can be wrong. The current error rate is lower and it does not depend on the testing dataset size. In real life prevalence of positives is much lower but the system doesn’t have a memory to store previous results and gain knowledge from tested images, therefore, it will not affect the efficiency of the system

Comment #4) The model comparison in Table 1 is really interesting but also frustrating.  Recall is (again) a useful concept in machine learning but not widely used in the diagnostics literature; the same goes for F-1 score. Either include definitions for these terms in the paper, or add a glossary in supplemental material.

Response: We agree with the reviewer that from the perspective of medical applications F-1 score and recall are not widely used matrics. Furthermore, F-1 score is directly proportional to accuracy; therefore to avoid repetition F-1 score column has been omitted from table 1. While accuracy and precision suggest that the model is suitable for detecting cancer, calculating recall reveals its weakness. Hence recall is necessary for that table 1.

Action in manuscript: F-1score column is omitted. The updated table is as below.

Model

Accuracy

(%)

Average precision (AP)

Recall

Training time(s)

Model Size (KB)

Total parameter

ROC-AUC(%)

MobileNetV2

85

89

87

1637.3

56,139

6,273,202

0.937

VGG-16

85

88

86

17048.34

76,385

16,321,458

0.94

InceptionV3

80

83

88

5272.85

1,24,095

25,080,722

0.908

InceptionResNetV2

84

84

87

8945.21

2,42,301

56,795,474

0.929

Xception

82.2

84

84

6391.65

157,050

27,285,146

0.51

MobileNet

88

90

88

1885.62

50,439

6,441,266

0.949

MobileNetV3Small

74

82

90

1521.64

7,074

1,596,642

0.829

MobilenetV3Large

76

79

86

1854.94

17,859

4,309,490

0.844

Squeeze-MNet(Proposed)

99.36

98

99

2271.60

50,439

6,441,266

0.989

Comment #5) The text on lines 216-218 needs to be unpacked/expanded a little bit to help readers understand how what they are looking at supports the claims that are made.  I was frustrated throughout by the claims that the model tuning approaches avoided over-fitting.  This is an important issue for out-of-sample accuracy but it's glossed over in the paper. We have to rely on the assumption that the approaches the authors mention are robust in avoiding over-fitting, but if we are not familiar with them then it's not easy to be sure.  A brief explanation of how optimization is achieved without over-fitting would be useful. It's a key point and worth explaining in some detail.

Response: The reviewer mentioned an important aspect of the paper. Here model optimization and learning curve performance are highly dependent on the head network as shown in figure 5. In the deep learning model training model, fine-tuning is achieved with the trial and error method. We need to check model performance with different parameters and select the best of all. There is not a single rule for all applications.

Action taken : (Text added on the 8th page)

The learning curve (Appendix A 2)shows that accuracy increased after each epoch because the model was fine-tuned(Appenxi A 3), and Leaky ReLU (Appendix A 4) activation after each dense layer prevented overfitting(Appendix A 5) and underfitting(Appendix A 6). The Leaky version of ReLU allows only a small gradient to pass. Figures 9a and b show the accuracy and loss by epoch. All of the optimization work has been done by the head network mounted on the MobileNet deep learning model as shown in figure 5. Hyperparameter tuning is also responsible for the robust model. It is explained in the 4.4 section.

Reviewer 2 Report

The merit of the proposed approach is supported by the results, but I miss on the paper a bit more discussion on why these techniques were chosen for this problem and had not been considered before. This however is more of a nitpicking than a detrimental comment.

The introduction is deprived of the related work with the recent literature.

It would be interesting if the authors report the trade-off compared to other methods especially the computational complexity of the models. Some techniques require more memory space and take longer time, please elaborate on that. 

Below papers has some interesting implications that you could discuss in your introduction and how it relates to your work.

S. Ali, et al. "Multitask Deep Learning for Cost-Effective Prediction of Patient's Length of Stay and Readmission State Using Multimodal Physical Activity Sensory Data," in IEEE Journal of Biomedical and Health Informatics, 2022, doi: 10.1109/JBHI.2022.3202178.

Subhan, F.; et al.. Cancerous Tumor Controlled Treatment Using Search Heuristic (GA)-Based Sliding Mode and Synergetic Controller. Cancers 2022, 14, 4191. https://doi.org/10.3390/cancers14174191

The introduction is not clear and very less literature is used. Follow these instruction: The introduction should briefly place the study in a broad context and highlight why it is important. It should define the purpose of the work and its significance, including specific hypotheses being tested. The current state of the research field should be reviewed carefully and key publications cited. Please highlight controversial and diverging hypotheses when necessary. Finally, briefly mention the main aim of the work and highlight the main conclusions. Keep the introduction comprehensible to scientists working outside the topic of the paper.

Authors should further clarify and elaborate novelty in their contribution.

What are the limitations of the present work?

Author Response

Reviewer 2
We would like to thank you reviewer for your valuable comments and suggestions. The answers to each comment are as given below.
--------------------------------------------------------------------------------------------------------------------- 
Comment #1: The merit of the proposed approach is supported by the results, but I miss on the paper a bit more discussion on why these techniques were chosen for this problem and had not been considered before. This however is more of a nitpicking than a detrimental comment.

Response:  This is a valuable comment therefore we included the objective of the research in the 1st paragraph of the introduction section. The proposed method is efficient and needs less technical skills for detection. Production cost is lower due to the use of IoT devices and it's fast. Our model is highly optimized to achieve great results.
Embedded devices have lower computing power and there is a need for low memory-consuming AI models to work with them with better accuracy. Other researchers proposed great models but detection time and memory requirements are higher. Therefore we used the selected pretrained model to reduce training time. For better feature extraction images have been preprocessed with a digital hair removal algorithm. This process reduced the computational cost of the model considerably.
To further enhance the novelty of this work more detailed text has been added on page 7.
Added text: 
In today's world, cancer is a deadly disease. It is the 3rd most common cause of death among humans, with a 78% death rate at later stages. Skin cancer is an abnormal growth of skin cells that develops in the body due to sunlight and UV rays [1]. It quickly invades nearby tissues and spreads to other body parts if not seen at earlier stages. Early diagnosis of skin cancer is a foundation to improve the outcomes and is correlated with 99% overall survival (OS) [2] [3]. It means there are higher chances of survival in the early stage. According to the Skin Cancer Foundation (SCF), there is an increase in skin cancer incidence globally [4]. More than 3 million cases will be detected worldwide in the year 2021.
Added text on page 7:
The training time factor affects the speed of the detection, when training time and total extracted features are higher the model is heavy (needs huge memory to run the model) in this case VGG-16 and Xception. When extracted features or training time is lower then the model gives less accuracy like MobilenetV3Small mode. The Proposed model gives the best tradeoff of size and accuracy
Machine learning also contributes to enhancing the mathematic prediction of cancer cell spreading rate[13]. Sajid Ali et. proposes the novel use of sensory data to predict the stay patient's length of stay in the hospital [14]. There are many deep learning models proposed by researchers but very few are suitable for IoT devices. Most AI model takes larger memory space and higher computational power for the best accuracy but our model has optimal complexity and the best accuracy

Comment #2: The introduction is deprived of the related work with the recent literature.

Response: We agree with the reviewer therefore we have added the objective and need for research is added in the 1st paragraph of the introduction section. Current research related to cancer with AI systems has been added from reference numbers 7 to 13. Limitations of current work have been studied and the solution is presented in our paper. 
Added text in the manuscript: 
The formal diagnosis method to detect cancer is visual inspection and biopsy. The primary visual examination includes the assistance of polarized light magnification via dermoscopy. A patient's history, social habits, skin color, occupation, ethnicity, and exposure to the sun are the critical factors considered during examinations. The laboratory biopsied the suspected lesion of concern. This method is painful, times consuming, and expensive for doctors and patients. Without insurance, a skin biopsy costs $10 to $1000 [6]. There is an urgent need for skin cancer detection based on Artificial Intelligence (AI) to overcome the above problems.
In terms of dermatology, various diagnostic models using medical images have been performed as well as clinicians [6]. Recently, deep learning has provided end-to-end solutions to detect COVID-19 infection, lung cancer, skin lesions, brain and breast tumors, stomach ulcers, and colon cancer, predict blood sugar levels and heart disease, and detect face masks [7-12]. Machine learning also contributes to enhancing the mathematic prediction of cancer cell spreading rate[13]. Sajid Ali et. proposes the novel use of sensory data to predict the stay patient's length of stay in the hospital [14]. There are many deep learning models proposed by researchers but very few are suitable for IoT devices. Most AI model takes larger memory space and higher computational power for best accuracy but our model has optimal complexity and better accuracy.

Comment #3) It would be interesting if the authors report the trade-off compared to other methods especially the computational complexity of the models. Some techniques require more memory space and take longer time, please elaborate on that. 

Response: The key contribution of our method is it is lightweight, fast, and accurate. The base mode MobileNet deep neural model. In table 1 we have compared other pretrained models with  MobileNetV1 and concluded that the proposed model's size is the lowest among all; it means it is lightweight. Column “Total parameter” in table 8 is features extracted from images to gain knowledge. Even though the MobileNet model is extracting fewer features than the VGG16 and InceptionV3 models but accuracy remains high. Therefore we concluded that the proposed model is best suitable for IoT devices. A detailed study has been added on page 7 of the manuscript.
Added text in the manuscript:
The training time factor affects the speed of the detection, when training time and total extracted features are higher the model is heavy (needs huge memory to run the model) in this case VGG-16 and Xception. When extracted features or training time is lower then the model is less accurate like MobilenetV3Small mode. The Proposed model gives the best tradeoff of size and accuracy.

Comment #4) Below papers has some interesting implications that you could discuss in your introduction and how it relates to your work.
S. Ali, et al. "Multitask Deep Learning for Cost-Effective Prediction of Patient's Length of Stay and Readmission State Using Multimodal Physical Activity Sensory Data," in IEEE Journal of Biomedical and Health Informatics, 2022, doi: 10.1109/JBHI.2022.3202178.
Subhan, F.; et al.. Cancerous Tumor Controlled Treatment Using Search Heuristic (GA)-Based Sliding Mode and Synergetic Controller. Cancers 2022, 14, 4191. https://doi.org/10.3390/cancers14174191

Response: As the reviewer suggested the above paper are showing the latest trend in healthcare. S. Ali mentions the use of sensory data to predict the stay of the patient in the hospital. It is the best suitable model for smart hospital and resource management. 2nd manuscript is about cancer cell treatment response with mathematical equations calculated for better predictions. The latest trend is an important aspect of any research study; therefore it is added in the introduction section.
Added text: The latest studies have been added in the 3rd paragraph.
Machine learning also contributes to enhancing the mathematic prediction of cancer cell spreading rate[8]. Sajid Ali et. proposes the novel use of sensory data to predict the stay patient's length of stay in the hospital [9].  

Comment #5) The introduction is not clear and very less literature is used. Follow these instructions: The introduction should briefly place the study in a broad context and highlight why it is important. It should define the purpose of the work and its significance, including specific hypotheses being tested. The current state of the research field should be reviewed carefully and key publications cited. Please highlight controversial and diverging hypotheses when necessary. Finally, briefly mention the main aim of the work and highlight the main conclusions. Keep the introduction comprehensible to scientists working outside the topic of the paper.

Response: As suggested by the reviewer the introduction has been improved and important text has been added. Study objectives and the need for research are in 1st paragraph of section 1. The latest study trend is cited from numbers 8 to 17. Limitations of the current method are also included in the 3rd paragraph. 
Added text in manuscript:
In today's world, cancer is a deadly disease. It is the 3rd most common cause of death among humans, with a 78% death rate at later stages. Skin cancer is an abnormal growth of skin cells that develops in the body due to sunlight and UV rays [1]. It quickly invades nearby tissues and spreads to other body parts if not seen at earlier stages. Early diagnosis of skin cancer is a foundation to improve the outcomes and is correlated with 99% overall survival (OS) [2] [3]. It means there are higher chances of survival in the early stage. According to the Skin Cancer Foundation (SCF), there is an increase in skin cancer incidence globally [4]. More than 3 million cases will be detected worldwide in the year 2021.
The formal diagnosis method to detect cancer is visual inspection and biopsy. The primary visual examination includes the assistance of polarized light magnification via dermoscopy. A patient's history, social habits, skin color, occupation, ethnicity, and exposure to the sun are the critical factors considered during examinations. The laboratory biopsied the suspected lesion of concern. This method is painful, times consuming, and expensive for doctors and patients. Without insurance, a skin biopsy costs $10 to $1000 [5]. There is an urgent need for skin cancer detection based on Artificial Intelligence (AI) to overcome the above problems.  
Embedded devices have lower computing power and there is a need for low memory-consuming AI models to work with them with better accuracy. Other researchers proposed great models but detection time and memory requirements are higher. Therefore we used the selected pretrained model to reduce training time. For better feature extraction images have been preprocessed with a digital hair removal algorithm. This process reduced the computational cost of the model considerably.
Machine learning also contributes to enhancing the mathematic prediction of cancer cell spreading rate[13]. Sajid Ali et. proposes the novel use of sensory data to predict the stay patient's length of stay in the hospital [14]. There are many deep learning models proposed by researchers but very few are suitable for IoT devices. Most AI model takes larger memory space and higher computational power for best accuracy but our model has optimal complexity and better accuracy.

Comment #6)  Authors should further clarify and elaborate novelty in their contribution.

Response: The novelty of our work is written in the 2nd paragraph of the 2nd page. Mainly this work is a digital hair removal algorithm for preprocessing images and an accurate classification model.  Each point is further elaborated in sections 4.1 and 4.2. The preprocessing algorithm gives rocket high accuracy. The before and after preprocessing learning curve proves the novelty of the work as shown in the below figure. Accuracy plot with each epoch is plotted 

Our method uses MobilenetV1 after the digital hair removal process for better feature extraction. It removes noise from the image and only important information is maintained in the image. The fabricated device is tested on a 64GB memory IoT device. It is a lightweight model with the best accuracy and precision. Accuracy, ROC value, false alarm, and confusion matrix values are supporting the model's efficiency.
To explain the novelty in the work more detail of the ROC curve and learning curve has been added in section 4.3 Squeeze-MNet model analysis.  
Action taken:  (Model performance is added in section 4)
The training time factor affects the speed of the detection, when training time and total extracted features are higher the model is heavy (needs huge memory to run the model) in this case VGG-16 and Xception. When extracted features or training time is lower then the model gives less accuracy like MobilenetV3Small mode. The Proposed model gives the best tradeoff of size and accuracy.
 In figure 8(a) red line is the performance of the model without knowledge and the blue line is the intelligence gained with each epoch. It is about gaining logic hence we name it a logistic skill. AUC is equivalent to the probability that a randomly chosen positive instance is ranked higher than a randomly chosen negative instance. The overall error rate (1- accuracy) is 0.64; It is a combination of false positive and false negative. Both values are mentioned in the confusion matrix in figure 8(b). Our system does not perform well with false positive values. In the medical field, false positive is stressful and false negative is fatal to patients. AI experts and researchers need to fix this issue.

Comment #7)  What are the limitations of the present work?

Response: Thank you for mentioning this important aspect of research. The major limitation of the system is the person operating the device must need previous knowledge to capture a clear picture. Device accuracy highly depends on the contrast, brightness, and sharpness of the image. As we mention in the manuscript we can not get permission to test it on real patient therefore device testing and optimization is necessary for practical application.
Added text in the manuscript:
A digital hair removal algorithm has to be used before object detection which can give a microsecond delay in detection. The major limitation of the system is specificity and sensitivity are still lower than accuracy. For practical application, it has to be higher than current values. If the model is trained with high definition image dataset we will overcome this issue. Another limitation is model only detects skin cancer but can not detect the type of cancer for further treatment. In future work, we are planning to include at least 25 different skin problems in a single model. 

Round 2

Reviewer 2 Report

.

Author Response

Dear Editor,

There is no text comment in the "Comments and the suggestions for Authors" section.

Therefore we do not have any response for the reviwer.